# OpenReview forum: "Endogenous Resistance to Activation Steering in Language Models"
_ICLR.cc/2026/Workshop/Sci4DL — Sci4DL 2026_

### Official Review · Reviewer_TNG9 · 2026-02-08

**Fit:** 3
**Significance:** 2
**Confidence:** 3

**Summary:**

This paper introduces Endogenous Steering Resistance (ESR), a phenomenon where language models recover mid-generation from activation steering while steering remains active. They employ SAE-based steering and ablations across multiple model scales to show that larger models (i.e. especially Llama-3.3-70B) exhibit higher ESR rates. They then identify candidate latents associated with this behavior, and demonstrate that fine-tuning increases self-correction attempts without improving correction success.

**Strengths:**

- Introduces and operationalizes a novel phenomenon with clear safety and robustness implications
- Includes thoughtful controls (no-steering baseline, random-latent ablations, steering sweeps)
- Fine-tuning provides a valuable caution against superficial “metacognitive” improvements

**Suggestions:**

- ESR definition hinges on explicit linguistic restart markers, confounding self-monitoring with verbalization style and likely inflating ESR in larger models (which would be interesting to see)
- Identified latents plausibly could encode surface hesitation/restart behavior rather than deeper off-topic detection
- Post-hoc layer selection, and modest effect sizes, and reliance on proprietary SAE assets may limit generality and reproducibility of their contribution

---

### Official Review · Reviewer_knxD · 2026-02-25

**Fit:** 3
**Significance:** 2
**Confidence:** 2

**Summary:**

This paper introduces Endogenous Steering Resistance (ESR), a phenomena where Large Language Models recover from task-misaligned activation steering using SAE latents during inference. The paper finds that Llama-3.3-70B reports a higher ESR (3.8%) than smaller models (<1%), suggesting stronger self-correction capabilities. Further, the paper finds a set of 26 Off-Topic Detector (OTD) SAE latents, that on ablation drop ESR (by ~25%) suggesting the presence of self-correction capabilities within the models. The use of meta-prompts for self-monitoring increases ESR significantly. Finally, finetuning Llama-3.1-8B on self-corrected responses increases the number of attempts to recover but does not necessarily result in improved responses.

**Strengths:**

- The causal analysis with the 26 identified OTD latents effectively establishes that they support self-monitoring.
- The fine-tuning experiment is also very interesting since it demonstrates that training the model on self-monitored responses only shows an increase in the number of attempts but not better quality responses.

**Suggestions:**

- While Llama-3.3-70B has a higher ESR than other smaller models, the ESR is still very low (3.8%) to indicate practical significance of this phenomenon. An analysis with more models of varying scales should help establish the validity of the claim.
- The 26 OTD latents are identified using differential analysis but it isn’t clear if this is a unique set. There could be a smaller set of SAE latents involved in self-correction and an ablation on this would be very useful.
- While the results indicate self-correction capabilities are tied to latents firing, its connection to self-correction “circuits” within the model is not established robustly.
- In the “Impacts for AI safety” section, the paper mentions that techniques such as Inference-Time Intervention (ITI) might be less successful if the model deems the steering irrelevant. There is, however, no clear analysis of when this might happen (what could be seen as irrelevant if the steering is task-specific?)

---

### Official Review · Reviewer_TYdE · 2026-02-25

**Fit:** 2
**Significance:** 1
**Confidence:** 3

**Summary:**

This paper identifies and characterises Endogenous Steering Resistance as a phenomenon, where LLMs spontaneously self-correct during inference when they are subjected to task-misaligned activation steering via SAE latents. By boosting off-topic SAE features during generation across five models, the authors find that the model generates off topic content but then explicitly interrupts itself and recovers. They identify 26 "off-topic detector" SAE latents via contrastive search where ablating them reduces ESR by 25%. Meta prompts instructing self-monitoring enhance ESR 4x in LLama-3.3-70B and fine-tuning on synthetic self-correction data induces the behavioural pattern but not the underlying improvement capability.

**Strengths:**

- ESR is a novel observation and phemonemon: sponatenous mid-generation self correction during activation steering is a new observation and understudied phenomenon. Regardless of effect size, identifying this behaviour is a contribution
- The experimental controls are good, the no steering baseline cleanly establishes that ESR is steering induced rather than spontaneous. The random latent ablation control confirms that the OTD ablation effect is specific to those latents. Validation with 5 different LLM judges (with high agreement) rules out evaluation artifacts to a large extent
- The finding that fine-tuning induces the behavioural pattern of self-correction without improving actual correction quality is an interesting insight about the nature of the learned behaviours

**Suggestions:**

- The 3.8% ESR rate does not support the main claim in the paper. The paper describes this as "substantially" higher than other models, but in absolute terms 96% of steered responses in the best performing model show no ESR. This is a rare event and not a robust phenomenon. The paper's narrative of emergent self monitoring circuits is built on very thin empirical ground
- The mechanistic analysis is lacking: out of the 26 identified OTD latents, approximately half have almost zero or negative Cohen's d values (in Table 2) which means that they activate more during on topic than off topic content. This directly contradicts their proposed role as off topic detectors. The contrastive search identified a heterogeneous set and only around half of them behave as predicted. Ablating all 26 reduces ESR by only 25%, leaving 75% unexplained.
- There is a confound in the scale vs architecture question which is unresolved. The 5 models cover 2 families and have different architecture, training and fine-tuning. The paper acknowledges that it cannot cleanly disentangle these factors, yet the framing strongly implies scale dependence. There would need to be at least one additional large model to support the scale emergence narrative
- The Goodfire SAE was trained on layer 50 but steering was applied at layer 33 which may introduce bias. This selection based on favourable outcomes could inflate observed ESR rates and should be acknowledged as a more serious limitation
- The definition of self-correction relies on explicit language and subtler resistance (e.g. gradual topic drift back to relevance) would be missed
- The paper does not explore how the OTD latents trigger self correction. What downstream circuits do they activate? How does the model transition from off topic to corrective generation?

---

### Meta-Review · Area_Chair_9Dc1 · 2026-03-01

**Recommendation:** Accept

**Metareview:**

While the paper makes important contributions that are worth sharing at the workshop, it could also benefit from several improvements as highlighted by Reviewer TYdE. I suggest the authors do so for the camera-ready version.

---

### Decision · Program_Chairs · 2026-03-02

Accept